# Patient Reported Outcome Measures of Sleep Quality in Fibromyalgia: A COSMIN Systematic Review

**DOI:** 10.3390/ijerph17092992

**Published:** 2020-04-26

**Authors:** Carolina Climent-Sanz, Anna Marco-Mitjavila, Roland Pastells-Peiró, Fran Valenzuela-Pascual, Joan Blanco-Blanco, Montserrat Gea-Sánchez

**Affiliations:** 1Faculty of Nursing and Physiotherapy, University of Lleida, 25198 Lleida, Spain; carol.climent@udl.cat (C.C.-S.); annamarco18@gmail.com (A.M.-M.); roland.pastells@udl.cat (R.P.-P.); fran.valenzuela@udl.cat (F.V.-P.); montse.gea@udl.cat (M.G.-S.); 2Grup d’Estudis Societat, Salut, Educació i Cultura, University of Lleida, 25001 Lleida, Spain; 3Grup de Recerca de Cures en Salut, Institut de Recerca Biomèdica de Lleida, IRB, 25198 Lleida, Spain

**Keywords:** patient-reported outcome measures, consensus-based standards for the selection of health measurement instruments, surveys and questionnaires, sleep quality, fibromyalgia, psychometrics, systematic review

## Abstract

Poor sleep quality is a common concern and a troublesome symptom among patients suffering from fibromyalgia. The purpose of this review was to identify and describe the available patient reported outcome measures (PROMs) of sleep quality validated in adult people diagnosed with fibromyalgia. The COSMIN and PRISMA recommendations were followed. An electronic systematized search in the electronic databases PubMed, Scopus, CINAHL Plus, PsycINFO, and ISI Web of Science was carried out. Validation studies of PROMs of sleep quality in fibromyalgia published in English or Spanish were included. The selection of the studies was developed through a peer review process through the online software “COVIDENCE”. The quality of the studies was assessed using the COSMIN Risk of Bias checklist. A total of 5 PROMs were found validated in patients with fibromyalgia: (1) Pittsburgh Sleep Quality Index (PSQI), (2) Jenkins Sleep Scale (JSS), (3) Sleep Quality Numeric Rating Scale (SQ-NRS), (4) Medical Outcomes Study-Sleep Scale (MOS-SS), and (5) Fibromyalgia Sleep Diary (FSD). The quality of the evidence was very good and the quality of the results ranged from moderate to high. All the included PROMs, except for the FSD, showed adequate psychometric properties and, therefore, are valid and reliable tools for assessing sleep quality in the context of FM. However, none of the studies analyzed all the psychometric properties of the included PROMs as established in the COSMIN guidelines, highlighting that this is a potential field of research for future investigations.

## 1. Introduction

Historically, fibromyalgia (FM) has been presented as a heterogeneous health condition and a multitude of symptoms associated with it have been described, which has made it difficult to establish the most prevalent and severe symptoms of this syndrome [1]. Likewise, as stated by Carmona et al. [2], FM is a challenging health condition given the lack of objective tests to monitor the evolution of the people suffering from it.

The OMERACT working group [3] established the central clinical domains that characterize FM using a Delphi study design that included physicians and patients. Their results showed a high level of agreement between professionals and patients that poor sleep quality is one of the main symptoms of FM. In addition, 92% of patients identified that the assessment of poor sleep quality should be carried out in all experimental studies about FM. These results are in line with a previous internet survey, including 2596 people with FM [4], showing that poor sleep quality, together with pain, fatigue and morning stiffness, are the symptoms with greater severity and impact in these patients. Specifically, 79% of the participants perceived that sleep problems were one of the most common factors in the exacerbation of FM symptoms [4].

In terms of prevalence, the studies indicate that between 65% and 99% of people diagnosed with FM report poor sleep quality [5,6,7]. Other authors [8] state that 63% of these patients report two or more symptoms of difficulty sleeping, while only 11.2% report having no problem sleeping. A recent meta-analysis of case-control studies indicated that, in comparison with healthy controls, people with FM show significantly lower sleep efficiency and sleep quality, shorter sleep duration, longer wake time after sleep onset and more percentage of light sleep stages when assessed with polysomnography. Subjective assessment showed that patients with FM have more difficulties falling asleep and worse sleep efficiency. Therefore, and although there are no conclusive data regarding the prevalence of poor sleep quality in FM, the results presented reveal that it is a recurrent and a concerning symptom among these patients [9]. Furthermore, poor sleep quality has been shown to be related with increases in the intensity of pain, and it is an aggravating factor of other FM symptoms such as fatigue, cognitive problems and quality of life [7,10]. 

Taking into account all these data, the assessment of sleep quality could guarantee comprehensive assistance in patients with FM and can provide important information on the effectiveness of prescribed treatments, both pharmacological and non-pharmacological [11]. In the field of research, the assessment of sleep quality may be of especial interest when evaluating the effectiveness of new treatments, and also to improve the knowledge on how this symptom can influence the general health status of people who suffer from FM [10]. 

Therefore, the main objective of this systematic review was to identify and describe the available PROMs of sleep quality in adults diagnosed with FM and their psychometric properties. In addition, adaptations and translations of these tools to other languages were also presented.

## 2. Materials and Methods

This review was carried out following the “COnsensus-based Standards for the selection of health Measurement Instruments” (COSMIN) [12] and the “Preferred Reporting Items for Systematic reviews and Meta-Analyses” (PRISMA) [13] guidelines.

The review protocol was registered with PROSPERO (Record ID = CRD42018114218). 

### 2.1. Criteria for Considering Studies for This Review

#### 2.1.1. Type of Studies

Validation or cross-cultural adaptation studies published in English or Spanish (the research team did not speak fluently other languages) with no restriction regarding the year of publication.

Additionally, the development studies for each of the included PROMs were searched so as to analyze the content validity for those PROMs originally developed in the context of FM. In the case of the PROMs that were developed for other target populations, the report was also searched and the results presented. 

#### 2.1.2. Type of Participants

Validation or cross-cultural adaptation studies involving adult participants (18 years or older) diagnosed with FM.

#### 2.1.3. Type of Outcome Measures

Studies that met the above inclusion criteria were included regardless of whether they did not report all the psychometric properties established in the COSMIN guidelines [12]: content validity, structural validity, internal consistency, cross-cultural validity, reliability, measurement error, criterion validity, hypotheses testing for construct validity, and responsiveness.

### 2.2. Search Strategy

An electronic systematized search in the electronic databases PubMed, Scopus, CINAHL Plus, PsycINFO, and ISI Web of Science was carried out.

The search strategy was developed by two authors (RP and AM) based on the COSMIN “search filters for finding studies on measurement properties” provided as an additional tool in the COSMIN website (https://www.cosmin.nl/tools/pubmed-search-filters/). The Peer Review of Electronic Search Strategies guidelines [14] recommendations were also implemented for the development of the search strategy in the selected databases. The following MeSH terms were used for the development of the search strategy: “Fibromyalgia”, “Sleep”, “Surveys and Questionnaires”, “Psychometrics”, and “Validation Studies as Topic”. Entry terms and free text terms derived from or related with each selected MeSH term were also included in the search strategy (Appendix A: “search strategy from the consulted databases”). 

The last search was run on March the 6th, 2020.

Searching other sources: a manual search of studies was carried out based on the bibliographic references of the included articles. 

### 2.3. Selection of Studies

The selection of studies process was developed with the online software “COVIDENCE”. The identified studies were first stored and checked for duplicates. After duplicates were removed and based on the inclusion criteria, two authors (CC and AM) carried out first the title and abstract screening, and subsequently the full-text screening of the studies through a peer review process. In case of discrepancy in any of the two phases of the selection of the studies, a third author (MG) discussed the suitability of the studies to be included.

### 2.4. Data Collection and Data Items

The data collection process was carried out by two authors (CC and AM) independently, and a third author (FV) reviewed the extraction so as to ensure accuracy of the data.

For each of the included PROMs, the following data items were extracted in accordance with the COSMIN recommendations [12]: characteristics of the included PROMs, characteristics of the included study populations, results of studies on measurement properties. 

### 2.5. Risk of Bias and Quality of the Results Assessment

Two authors (CC and FV) rated independently the RoB of each of the included studies and the quality of the results following the COSMIN RoB checklist [15]. A third author (MG) intervened in case of discrepancy. The COSMIN RoB is comprised by ten checklists evaluating the following methodological aspects: (1) PROM development, (2) Content validity, (3) Structural validity, (4) Internal consistency, (5) Cross-cultural validity/Measurement invariance, (6) Reliability, (7) Measurement error, (8) Criterion validity, (9) Hypotheses testing for construct validity, and (10) Responsiveness. Each of the checklists includes different items that can be rated as “very good”, “adequate”, “doubtful”, “inadequate”, and “not applicable”. An excel document is provided on the COSMIN website to facilitate the RoB assessment (https://www.cosmin.nl/tools/guideline-conducting-systematic-review-outcome-measures/?portfolioCats=19). 

The quality of the results was evaluated after the data extraction regarding the measurement properties of each of the included PROM in accordance with the COSMIN pre-established criteria [12,16] (more details in Table 1).

### 2.6. Data Analysis and Synthesis of Results

A narrative synthesis of the results was carried out. 

## 3. Results

### 3.1. Study Selection

The electronic literature search yielded 3042 records in total and 1410 duplicates were removed. During the process of study selection, 1632 records were analyzed by title and abstract, and 1620 were excluded. Finally, 12 records were selected for the full text analysis, and 6 studies met the inclusion criteria. The manual search based on the bibliographic references of the included studies yielded one study. Therefore, seven studies were included in the narrative synthesis and five instruments were described (Figure 1: Process of Study Selection (PRISMA Flow Diagram) [13].)

### 3.2. Risk of Bias

All the included studies [17,18,19,20,21,22,23] showed a very good methodological quality for assessing the measurement properties of the selected PROMs in accordance with the COSMIN criteria. Therefore, the RoB was rated as low for all the studies. The results from the risk of bias assessment can be consulted in Table 2.

### 3.3. Characteristics of the Included PROMs and the Study Populations

The characteristics of the included PROMs and the characteristics of the study populations can be consulted in Table 3 and Table 4, respectively. 

The included studies reported the following PROMs for sleep quality in patients with FM: (1) Pittsburgh Sleep Quality Index, (2) Jenkins Sleep Scale (alternative scoring method), (3) Sleep Quality-Numeric Rating Scale, (4) Medical Outcomes Study-Sleep Scale, and (5) Fibromyalgia Sleep Diary.

For those of the included PROMs that were not originally developed as specific tools for assessing sleep quality in patients with FM, the characteristics of the PROM and the study populations of the original development studies were also included in Table 3 and Table 4. 

#### 3.3.1. Pittsburgh Sleep Quality Index (PSQI)

The PSQI was developed with the understanding that the essential elements that characterize good sleep are mainly subjective and may vary between individuals. Accordingly, Buysse et al. [24], pointing out that poor sleep quality was a highly prevalent problem in people with psychiatric problems, developed the first version of this index in 1989 with the objective of assessing in a reliable and valid way the quality of sleep from the perspective of patients.

Objective of the tool [24]: The PSQI assesses the sleep quality of the month prior to the evaluation since, as the authors stated in the “Consensus Conference of Insomnia, 1984,” it was established that the assessment of 2–3 weeks of sleep is the ideal minimum time for being able to discern between transient and persistent sleep problems. Accordingly, the PSQI allows the latter distinction to be made if it is applicable twice with one month of separation. 

Number of items and response options [24]: To assess the described components, the PSQI is composed of 19 self-rated questions and 5 questions that are answered by the roommate or bedmate, although the latter are only used for clinical purposes and are not included in the final score. The items of the PSQI are organized into seven components: (1) Subjective sleep quality, (2) Sleep latency, (3) Sleep duration, (4) Habitual sleep efficiency, (5) Sleep disturbances, (6) Use of sleeping medication, and (7) Daytime dysfunction. The first four items are answered by providing some data related to the usual time of sleep, time to fall asleep, time awake at night, and hours of sleep per night. The other 12 items to be filled out by the patient plus the items to be filled in by the roommate or bed partner use the previous set of answers, and the respondent is asked to mark an X for the option that most corresponds to their experience. There are four possible answers, for example: (1) Not during the past month, (2) Less than once a week, (3) Once or twice a week, or (4) Three or more times a week. In one of the items, the possible answers are: (1) Very good, (2) Fairly good, (3) Fairly bad or (4) Very bad. Another item has the following answers: (1) No problem at all, (2) Only a very slight problem, (3) Somewhat of a problem, or (4) A very big problem. Finally, the last question has also four possible answers: (1) No bed partner or roommate, (2) Partner/roommate in other room, (3) Partner in the same room, but not same bed, or (4) Partner in the same bed.

Administration method and time of response [24]: The PSQI is self-completed by the respondent, and it takes 5–10 min to complete the questionnaire. 

Scoring: The total score of the questionnaire is derived from the sum of the seven components of the questionnaire. Each of the items has, as explained above, four possible answers, so the scoring varies between 0 and 3. In this way, the maximum final score is 21 points and the minimum score is 0.

Score interpretation [24]: According to the authors, a score lower than 5 points would indicate that the respondent is a “good sleeper” while ratings greater than 5 points would be indicative of poor sleep quality and moderate difficulties in three components or serious difficulties in at least two components of the seven that are evaluated.

Method of development [24]: The PSQI was developed based on the clinical experience of the authors with patients with sleep disorders and the results of a review of the previous literature through which the authors identified the already developed tools for the assessment of sleep quality. This process ensured that prior to the development of the PSQI, there were already different sleep measurement tools. However, there were very few that had been developed with clinical subjects. In addition, the PSQI allows the assessment of the sleep quality of the previous month, unlike other scales that only permit the assessment of the previous night or sleep problems during the year prior to the evaluation.

The authors defined four objectives [24]: (1) to develop a standardized, reliable, and valid tool to assess the quality of sleep, (2) to differentiate good and bad sleepers, (3) to develop an easy-to-complete and easy-to-interpret tool, and (4) to develop a short and clinically useful tool to assess a series of sleep problems that can interfere with the quality of it.

For the development of the PSQI, the authors recruited a sample composed of 52 healthy subjects defined as “good sleepers”, who formed control group I; 34 patients admitted to or outpatients of a psychiatric center diagnosed with major depressive disorder and considered “bad sleepers” who formed group II; and finally, 62 patients with sleep disorders referred by a physician from another psychiatric center formed group III.

After the development of the PSQI, an 18-month field testing was carried out to assess the clinical experience of its use.

Translations/adaptations in patients with FM: According to the results of our review, the PSQI was also validated in a sample of people diagnosed with FM in Spain [17] (more details in Table 3 and Table 4). 

#### 3.3.2. Jenkins Sleep Scale (JSS)

The development of the JSS was based on the absence of brief and easy-to-use sleep evaluation scales in the field of epidemiological research. In addition, the available tools only allowed the assessment of very specific sleep conditions. Thus, the main objective of the JSS was not to serve as a tool for assessing specific sleep problems such as narcolepsy or sleep apnea, but to allow evaluation of the most common symptoms in the general population [25].

Objective of the tool [25]: The JSS permits the assessment of the most common symptoms of insomnia (difficulty falling asleep and maintaining sleep, as well as the sensation of fatigue upon awakening) during the previous month.

Number of items and response options [25]: The scale consists of four items: (1) Do you have trouble falling sleep? (2) Do you wake up several times per night? (3) Do you have trouble staying asleep? (including waking far too early), and (4) Do you wake up after your usual amount of sleep feeling tired and worn out?

Each of the items is classified on a Likert scale of 6 points based on the frequency with which the respondent experiences each of the evaluated symptoms (0 = not at all, 1 = 1–3 days, 2 = 4–7 days, 3 = 8–14 days, 4 = 15–21 days, and 5 = 22–31 days). 

Administration method and time of response [25]: The JSS is a self-administered, brief, and quick-filling scale.

Scoring: According to the response options previously presented, the results of the JSS can vary from 0 to 20 in the total sum of the items.

Score interpretation [25]: 0 points are indicative that there are no sleep problems and 20 points indicate significant sleep problems.

Method of development [25]: The scale was developed within the framework of two other larger projects, the “Air Traffic Controller (ATC) Health Change Study” and the “Recovery Study” (RS). In the former, 300 questionnaires were sent by post, of which 250 were completed and returned. The sample consisted mainly of men between 25 and 49 years of age and the average age of respondents was 37.1 years. In the case of the RS, 467 subjects admitted for cardiac valve surgery or coronary bypass were included. A total of 80% of the sample consisted of white men between the age of 25–69 years, although most were between 50 and 60 years old.

Although the JSS was used for both studies, in the ATC study, all four items that make up the scale were included, but in the RS the item “waking up several times per night” was omitted, and the last two categories of responses were also modified and grouped into one, so that the last answer option was “15–31 days” and the total score could range between 0 and 12 points.

In the ATC study, the scale was administered only once, while in the RS study, it was completed before the surgery and at 6 and 12 months after the same.

Translations/adaptations in patients with FM: Crawford et al. [18] validated a JSS version with an alternative method of scoring in a sample of patients diagnosed with FM. The authors hypothesized that using a scoring method in which the patients must recall the exact number of nights they had sleep disturbance could increase the likelihood of incurring a recall bias. Therefore, an alternative scoring method was proposed in which the respondent must select a period of time instead of an exact number of days: (1) not at all (score = 0), (2) less than half the time (score = 1), and (3) greater than half the time (score = 2). Hence, the total score ranges from 0 to 8, higher scores being indicative of greater severity of sleep problems (more details in Table 3 and Table 4).

#### 3.3.3. Sleep Quality-Numeric Rating Scale (SQ-NRS)

The Sleep SQ-NRS was developed in order to collect relevant and appropriate information for a generic approach to the global impact of sleep problems in patients with FM [19]. 

Objective of the tool: The NRS is eligible in evaluations that require a daily record of the quality of sleep, offering the patient an element with little time burden.

Number of items and response options [19]: It is a tool with a single element. The patient is asked to choose the one that best describes their sleep quality during the last 24 h on a numerical scale of 11 points (0–10).

Administration method and time of response [19]: Self-managing scale and quick response. The patient is instructed to complete the tool just after waking up.

Scoring and score interpretation: The scoring scale fluctuates in a range between 0 “best possible sleep” and 10 “worst possible sleep”.

Translations/adaptations in patients with FM: no validations were found in other languages.

#### 3.3.4. Medical Outcomes Study-Sleep Scale (MOS-SS)

The development of the MOS-SS was derived from the results of a larger research project called the Medical Outcomes Study (MOS) [27], which consisted of a longitudinal descriptive observational study linked to the health outcomes in patients with chronic diseases. In this study, the authors concluded that sleep is a key factor for the functionality and well-being of people with chronic health conditions. In addition, the authors stated that sleep assessment could be key to understanding the health problems associated with chronic health conditions and developing more effective treatments.

Objective of the tool: The MOS-SS allows the assessment of sleep quality.

Number of items and response options [1,26]: The MOS-SS is composed of 12 items that evaluate 6 sleep domains: initiation (time to fall asleep), quantity (hours of sleep each night), maintenance, respiratory problems, perceived adequacy, and drowsiness.

The sleep scale uses a wide variety of response sets. The first item: (1) 0–15 min, (2) 16–30 min, (3) 31–45 min, (4) 46–60 min, (5) More than 60 min. The second item is an open question allowing a response that ranges from 0–24 h. The remaining 10 items use a set of 6-point answers based on the following values: (1) All of the time, (2) Most of the time, (3) A good bit of the time, (4) Some of the time, (5) A little of the time, (6) None of the time.

There is a nine-item version of the MOS-SS, named Sleep Problems Index II, and a 6-item version defined as the Sleep Problems Index I. Neither of the scales excludes any item from the original scale, but rather groups them in unique items so that the only difference is that these two scales are shorter.

Administration method and time of response [20]: The MOS-SS is a self-administered scale and takes about 2–3 min to complete.

Scoring [19]: Each of the response options described above is accompanied by a numerical index. The sum of the scores of the items and domains becomes a numerical scale of 0–100 in all the items. Two exceptions are contemplated: the score of the item “quantity” ranges between 0–24 and the score of the item “adequacy of sleep” ranges between 0–1. Regarding the interpretation of the score, higher scores indicate greater affectation of the variable that is being measured. In relation to sleep maintenance, it is considered optimal if the patient reports 7–8 h of sleep, assessed as 1, otherwise the score is 0.

Score interpretation [1]: According to the authors, high scores indicate worse sleep problems. The exceptions are the items “sufficiency of sleep” and “quantity” where lower scores indicate worse sleep problems.

Translations/adaptations in patients with FM: According to the results of this systematic review, there are three studies evaluating the content validity [19], the psychometric properties [20] and the test–retest reliability [21] of the MOS-SS in patients with FM (more details in Table 3 and Table 4).

#### 3.3.5. Fibromyalgia Sleep Diary (FSD)

Objective of the tool: In contrast with the above presented PROMs, the FSD was the first tool originally developed for evaluating sleep quality in people diagnosed with FM on a daily basis [23]. 

Number of items and response options [23]: The FSD is composed of eight items: (1) How difficult was it to fall asleep last night? (2) How restless was your sleep last night?, (3) How difficult was it to get comfortable last night?, (4) How difficult was it to stay asleep last night?, (5) How deep was your sleep last night?, (6) How rested were you when you woke up for the day?, (7) How difficult was it to begin your day?, (8) Did you have enough sleep last night?. The response options are based on a numerical scale of 11 points ranging from 0 to 10.

Score and score interpretation [23]: not reported. 

Method of development [23]: The FSD was developed by Kleinman et al. [23] in 2014 through a multi-staged process including a review of the literature, different qualitative approaches with experts and patients such as semi-structured interviews and focus groups, development of the conceptual framework and the first version of the FSD, and cognitive interviews so as to analyze the content validity and comprehensiveness of the PROM. The psychometric properties of the FSD were not analyzed.

For the development of the items, the authors used the terminology that emerged from the focus groups with the patients so as to ensure the adequacy of the content to people diagnosed with FM.

Translations/adaptations in patients with FM: no validations were found in other languages.

### 3.4. Results of Studies on the Measurement Properties in People Diagnosed with FM

None of the included studies reported all the measurement properties established by the COSMIN guidelines [12].

For the PSQI, the authors [17] reported data regarding internal consistency, reliability and hypotheses testing and were rated as positive (more details in Table 5: Results of studies on measurement properties (PSQI)).

The reported measurement properties for the JSS with an alternative scoring method [18] were internal consistency, criterion validity, test–retest reliability, and responsiveness. The results from internal consistency and responsiveness were rated as positive, while structural validity, criterion validity and reliability obtained moderate quality of the results, as some of them did not achieve the minimum standards established by the COSMIN guidelines [12] (more details in Table 6. Results of studies on measurement properties (JSS)).

The SQ-NRS content validity was evaluated by Martin et al. [19], showing favorable results for patients with FM. Cappelleri et al. [22] analyzed the criterion validity, the test–retest reliability and the responsiveness of the SQ-NRS showing positive results (more detail in Table 7. Results of studies on measurement properties (SQ-NRS)).

Regarding the MOS-SS, the results obtained by Martin et al. [19] provided strong evidence for validating the content of the tool in people diagnosed with FM. Cappelleri et al. [20] reported the structural validity and the internal consistency of the MOSS-SS, while the 1-week reliability of the scale was assessed by Sadosky et al. [21]. The results were positive for the internal consistency and for the reliability. However, the structural validity was rated as negative (more details in Table 8. Results of studies on measurement properties (MOS-SS)).

Additionally, Cappelleri et al. [20] estimated that a change of 7.9 of the total score represents the minimal clinical important change of the MOS-SS.

In relation to the FSD, the included study [23] aimed to develop and to analyze the content validity of the tool through a qualitative approach. The qualitative results showed that the FSD strongly represents the elements of sleep quality as a construct in the context of FM. As the psychometric evaluation of the FSD was not performed there were no statistical data to summarize.

## 4. Discussion

This systematic review aimed to describe the available PROMs for assessing sleep quality in people diagnosed with FM and to present and analyze their psychometric properties. A total of seven studies [17,18,19,20,21,22,23] and five PROMs were included in this systematic review: (1) Pittsburgh Sleep Quality Index, (2) Jenkins Sleep Scale, (3) Sleep Quality Numeric Rating Scale, (4) Medical Outcomes Study-Sleep Scale, and (5) Fibromyalgia Sleep Diary. All of the included studies presented low RoB for the analyzed psychometric properties according to the COSMIN RoB checklist, indicating very good methodological quality [15]. Likewise, the quality of the results ranged from moderate to high in accordance with the established COSMIN standards [12]. Although not all of the included studies conducted an analysis of the content validity of the PROMs, this systematic review found that the concept of sleep quality in the context of FM is homogeneous across included studies.

The PSQI showed high quality results for internal consistency, test–retest reliability, and hypotheses testing. For the JSS, the results were high quality for internal consistency and responsiveness and moderate for criterion validity and test–retest reliability as some of the items analyzed did not achieve the minimum pre-established standards. The SQ-NRS showed high quality for content validity, criterion validity, test–retest reliability, and responsiveness. For the MOS-SS, the results for structural validity, content validity, internal consistency, and test–retest reliability were rated as high quality. With regard to the FSD, the authors analyzed only the content validity, which demonstrated high quality results. Therefore, the PSQI, the JSS, the SQ-NRS, and the MOS-SS present satisfactory psychometric properties and are valid and reliable tools for assessing sleep quality in the context of FM.

Interestingly, the FSD is the only PROM specifically developed for evaluating sleep quality in patients with FM and is also the only one in which the psychometric properties were not analyzed. The latter highlights the need for future studies investigating if the FSD is a valid and reliable PROM in the context of FM.

In the context of clinical practice, the SQ-NRS is likely the most adequate measure for a rapid visual analogue scale-format evaluation of global severity of poor sleep quality, due to time restraints that usually accompany healthcare practice [28,29]. However, because the subjective perception of poor sleep quality in people diagnosed with FM is associated with alterations to different aspects of sleep (e.g., problems falling and staying asleep), tools allowing for a more comprehensive assessment of those sleep aspects could provide valuable information on how poor sleep quality impacts the general health of patients with FM and guide the development of more individualized treatment approaches. The PSQI, the JSS, and the MOS-SS permit the assessment of various components of sleep during the month prior to their completion, which provides concrete information on those aspects of sleep that are most affected.

At the research level, using valid and reliable PROMs for sleep quality in the context of FM could improve the quality of the studies’ results and increase knowledge of the relationship between poor sleep quality and other FM symptoms. Moreover, when investigating new treatment approaches, using PROMs for sleep quality that have been validated in people with FM could provide more reliable conclusions about their effectiveness [9]. Although the SQ-NRS is as valid and reliable as the other included PROMs, using tools such as the PSQI, the JSS, and the MOS-SS which permit a more comprehensive evaluation of sleep quality, could provide more accurate information on the effects of new interventions on specific sleep quality aspects and how these aspects relate with other FM symptoms. In this regard, the PSQI is the most widely used PROM among the existing literature in the field of FM, providing relevant information about both the relationship of poor sleep quality to other symptoms of this health condition [30,31,32,33] and the effects of different treatment approaches [34,35,36,37].

## 5. Conclusions

In conclusion, this systematic review revealed that the available PROMs for assessing sleep quality in people diagnosed with FM are valid and reliable. However, this subject remains a vital field of research as none of the included studies reported the complete list of psychometric properties established in the COSMIN guidelines [12]. In particular, the FSD, which is the only PROM specifically developed for people diagnosed with FM, should be analyzed for its validity and reliability.

## Figures and Tables

**Figure 1 ijerph-17-02992-f001:**
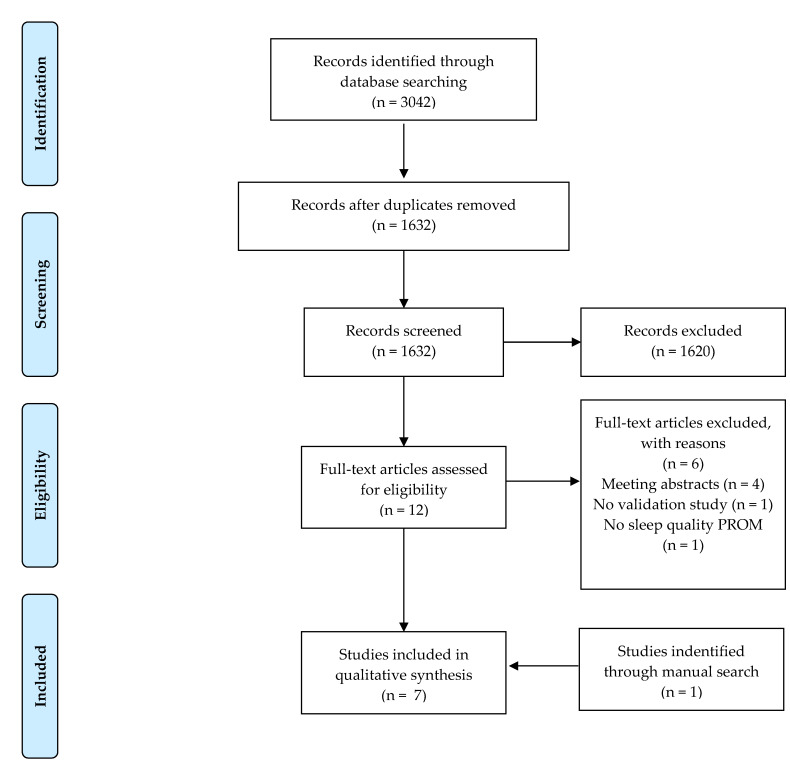
Process of Studies Selection (PRISMA Flow Diagram).

**Table 1 ijerph-17-02992-t001:** Criteria for evaluation of the quality of results.

Measurement Property	Rating	Criteria
Structural Validity	+	CTT CFA: CFI or TLI or comparable measure >0.95 OR RMSEA < 0.06 OR SRMR < 0.08 ^a^ IRT/Rasch No violation of unidimensionality ^b^: CFI or TLI or comparable measure > 0.95 OR RMSEA < 0.06 OR SRMR < 0.08 AND no violation of local independence: residual correlations among the items after controlling for the dominant factor < 0.20 OR Q3s < 0.37 AND no violation of monotonicity: adequate looking graphs OR item scalability > 0.30 AND adequate model fit IRT: χ2 > 0.001 Rasch: infit and outfit mean squares ≥ 0.5 and ≤ 1.5 OR Z-standardized values > −2 and < 2
	?	CTT: not all information for ‘+’ reported IRT/Rasch: model fit not reported
	−	Criteria for ‘+’ not met
Internal Consistency	+	At least low evidence ^c^ for sufficient structural validity ^d^ AND Cronbach’s alpha(s) ≥ 0.70 for each unidimensional scale or subscale ^e^
	?	Criteria for “At least low evidence ^c^ for sufficient structural validity ^d^” not met
	−	At least low evidence ^c^ for sufficient structural validity ^d^ AND Cronbach’s alpha(s) < 0.70 for each unidimensional scale or subscale ^e^
Reliability	+	ICC or weighted Kappa ≥ 0.70
	?	ICC or weighted Kappa not reported
	−	ICC or weighted Kappa < 0.70
Measurement Error	+	SDC or LoA < MIC ^d^
	?	MIC not defined
	−	SDC or LoA > MIC ^d^
Hypotheses Testing for Construct Validity	+	The result is in accordance with the hypothesis ^f^
	?	No hypothesis defined (by the review team)
	−	The result is not in accordance with the hypothesis ^f^
Cross-Cultural Validity/Measurement Invariance	+	No important differences found between group factors (such as age, gender, language) in multiple group factor analysis OR no important DIF for group factors (McFadden’s R2 < 0.02)
	?	No multiple group factor analysis OR DIF analysis performed
	−	Important differences between group factors OR DIF was found
Criterion Validity	+	Correlation with gold standard ≥ 0.70 OR AUC ≥ 0.70
	?	Not all information for ‘+’ reported
	−	Correlation with gold standard < 0.70 OR AUC < 0.70
Responsiveness	+	The result is in accordance with the hypothesis ^f^ OR AUC ≥ 0.70
	?	No hypothesis defined (by the review team)
	−	The result is not in accordance with the hypothesis ^f^ OR AUC < 0.70

Developed by Abedi, Prinsen, Shah, Buser and Wang [16], based on Prinsen et al. [12] under a Creative Commons Attribution 4.0 International License (http://creativecommons.org/licenses/by/4.0/). *AUC* area under the curve, *CFA* confirmatory factor analysis, *CFI* comparative fit index, *CTT* classical test theory, *DIF* differential item functioning, *ICC* intraclass correlation coefficient, *IRT* item response theory, *LoA* limits of agreement, *MIC* minimal important change, *RMSEA* root mean square error of approximation, *SEM* standard error of measurement, *SDC* smallest detectable change, *SRMR* standardized root mean residuals, *TLI* Tucker–Lewis index, *+* sufficient, − insufficient, ? indeterminate. ^a^ To rate the quality of the summary score, the factor structures should be equal across studies; ^b^ Unidimensionality refers to a factor analysis per subscale, while structural validity refers to a factor analysis of a (multidimensional) patient-reported outcome measure; ^c^ As defined by grading the evidence according to the GRADE approach; ^d^ This evidence may come from different studies; ^e^ The criteria “Cronbach alpha < 0.95” was deleted, as this is relevant in the development phase of a PROM and not when evaluating an existing PROM; ^f^ The results of all studies should be taken together, and it should then be decided if 75% of the results are in accordance with the hypotheses.

**Table 2 ijerph-17-02992-t002:** Risk of Bias assessment.

PROM	Measurement Properties Assessed	Risk of Bias
PSQI	Internal Consistency	Low
	Reliability	Low
	Structural validity	Low
	Hypothesis testing	Low
JSS	Internal Consistency	Low
	Reliability	Low
	Structural validity	Low
	Responsiveness	Low
SQ-NRS	Content validity	Low
	Reliability	Low
	Hypothesis testing	Low
MOS-SS	Content validity	Low
	Internal Consistency	Low
	Reliability	Low
FSD	Content validity	Low

PSQI: Pittsburgh Sleep Quality Index, JSS: Jenkins Sleep Scale, SQ-NRS: Sleep Quality-Numeric Rating Scale, MOS-SS: Medical Outcomes Study-Sleep Scale, FSD: Fibromyalgia Sleep Diary.

**Table 3 ijerph-17-02992-t003:** Characteristics of the included PROMs.

PROM * (Reference to First Article)	Construct(s)	Target Population	Mode of Administration (e.g., Self-Report, Interview-Based, Parent/Proxy Report etc.)	Recall Period	(Sub)scale(s) (Number of Items)	Response Options	Range of Scores/Scoring	Original Language	Available Translations
Pittsburgh Sleep Quality Index [24]	Sleep Quality	Patients diagnosed with major depressive disorder	Self-completed by the respondent	One month	Subscales: (1) subjective sleep quality, (2) sleep latency, (3) sleep duration, (4) habitual sleep efficiency, (5) sleep disturbances, (6) use of sleeping medication, and (7) daytime dysfunction. Items: 19 self-rated questions and 5 questions that are answered by the roommate or bedmate	The first four items are answered by providing some data related to the usual time of sleep, time to fall asleep, time awake at night, and hours of sleep per night. The other 12 items to be filled out by the patient plus the items to be filled in by the roommate or bed partner use the previous set of answers, and the respondent is asked to mark an X for the option that most corresponds to their experience: (1) Not during the past month, (2) Less than once a week, (3) Once or twice a week, or (4) Three or more time a week. In one of the items the possible answers are: (1) Very good, (2) Fairly good, (3) Fairly bad or (4) Very bad. Another item has the following answers: (1) No problem at all, (2) Only a very slight problem, (3) Somewhat of a problem, or (4) A very big problem. Finally, the last question has also four possible answers: (1) No bed partner or roommate, (2) Partner/roommate in other room, (3) Partner in the same room, but not same bed, or (4) Partner in the same bed.	The total score of the questionnaire is derived from the sum of the seven components of the questionnaire. Each of the items has, as explained above, four possible answers, so the scoring varies between 0 and 3. In this way, the maximum final score is 21 points and the minimum score is 0. a score lower than 5 points would indicate that the respondent is a “good sleeper” while ratings greater than 5 points would be indicative of poor sleep quality and moderate difficulties in three components or serious difficulties in at least two components of the seven that are evaluated.	English	Spanish with a sample of people diagnosed with FM [17]
Jenkins Sleep Scale [25]	Symptoms of insomnia	Patients 6 months after cardiac surgery Air traffic controllers	Self-administered	One month	The scale consists of four items: (1) Do you have trouble falling sleep? (2) Do you wake up several times per night? (3) Do you have trouble staying asleep? (Including waking far too early), and (4) Do you wake up after your usual amount of sleep feeling tired and worn out?	Each of the items is classified on a Likert scale of 6 points based on the frequency with which the respondent experiences each of the evaluated symptoms (0 = not at all, 1 = 1–3 days, 2 = 4–7 days, 3 = 8–14 days, 4 = 15–21 days, and 5 = 22–31 days).	According to the response options previously presented, the results of the JSS can vary from 0 to 20 in the total sum of the items. 0 points are indicative that there are no sleep problems and 20 points indicate significant sleep problems.	English	An alternative scoring method for the JSS was validated in Spanish with a sample of people diagnosed with FM [18].
Sleep Quality Numeric Rating Scale [19]	Sleep Quality	People diagnosed with FM	Self-administered	Daily record of the quality of sleep	It is a tool with a single element. The patient is asked to choose the one that best describes their sleep quality during the last 24 h on a numerical scale.	A numerical scale of 11 points (0–10).	The scoring scale fluctuates in a range between 0 “best possible sleep” and 10 “worst possible sleep”.	English	-
Medical Outcomes Study-Sleep Scale [1,19,20,26]	Sleep quality and quantity	Healthy adults and adults diagnosed with neuropathic pain	Self-administered	One month	The MOS-SS is composed of 12 items that evaluate six sleep domains: initiation (time to fall asleep), quantity (hours of sleep each night), maintenance, respiratory problems, perceived adequacy, and drowsiness.	The first item: (1) 0–15 min, (2) 16–30 min, (3) 31–45 min, (4) 46–60 min, (5) More than 60 min. The second item is an open question allowing a response that ranges from 0–24 h. The remaining ten items use a set of 6-point answers based on the following values: (1) All of the time, (2) Most of the time, (3) A good bit of the time, (4) Some of the time, (5) A little of the time, (6) None of the time.	According to the authors, high scores indicate worse sleep problems. The exceptions are the items “sufficiency of sleep” and “quantity” where lower scores indicate worse sleep problems.	English	English with a sample of people diagnosed with FM [19,20,21]
Fibromyalgia Sleep Diary [23]	Sleep quality	People diagnosed with FM	Self-administered	Daily record of the quality of sleep	The FSD consist of eight items: (1) How difficult was it to fall asleep last night?, (2) How restless was your sleep last night?, (3) How difficult was it to get comfortable last night?, (4) How difficult was it to stay asleep last night?, (5) How deep was your sleep last night?, (6) How rested were you when you woke up for the day?, (7) How difficult was it to begin your day?, and (8) Did you have enough sleep last night?	A visual analogue scale of 11 points ranging from 0 to 10.	Not provided	English	-

* Patient-Reported Outcome Measure.

**Table 4 ijerph-17-02992-t004:** Characteristics of the study populations.

		Population			Disease Characteristics			Instrument Administration			
PROM	Ref	N	Age Mean (SD, Range) yr	Gender % Female	Disease	Disease Duration Mean (SD) yr	Disease Severity	Setting	Country	Language	Response Rate
Pittsburgh Sleep Quality Index	24	Sample 1 = 34 Sample 2 = 45 Sample 3 = 17 Sample 4 = 52	Sample 1: 50.9 (range: 21–80) Sample 2: 44.8 (range: 20–80) Sample 3: 42.2 (range: 19–57) Sample 4: 59.9 (range: 24–83)	Sample 1: 26.4% Sample 2: 64.4% Sample 3: 52.9% Sample 4: 23.07%	Sample 1: Major depressive disorder Sample 2: Disorder of Initiating and Maintaining Sleep Sample 3: Disorders of Excessive Somnolence Sample 4: Healthy subjects	-	-	Psychiatric Clinics	United States of America	English	93.67%
	17	138	52.83 (±9.32)	100% women	Fibromyalgia	15.77 years (± 9.76)	Moderate: FIQ < 70 N = 68 FIQ score (51.02 ± 16.28) Severe: FIQ ≥ 70 N = 70 FIQ score (80.44 ± 6.20)	Community (FM association)	Spain	Spanish	Test: 100% Retest: 69.56%
Jenkins Sleep Scale	25	Sample 1 = 300 Sample 2 = 467	Sample 1: 37.1 (25–49) Sample 2: 54.9 (25–69)	Sample 1: 0% Sample 2: 20%	Sample 1: Air Traffic Controllers Sample 2: Cardiac valve surgery or coronary bypass	Sample 1: - Sample 2: -	Sample 1: - Sample 2; -	Sample 1: community Sample 2: Secondary health care	United States of America	English	Sample 1: 83.33% Sample 2: Test: 100% Retest: 91.22%
	18	195	46.5 (±11.35)	94.4%	Fibromyalgia	∼9 years	-	Clinical setting (unspecified)	United States of America	English	97.95%
Sleep Quality Numeric Rating Scale	20	Sample 1 = 748 Sample 2 = 745	Sample 1: 48.8 (±10.9) Sample 2: 50.1 (±11.4)	Sample 1: 94.4% Sample 2: 94.5%	Fibromyalgia	Sample 1: ∼9 years Sample 2: ∼10 years	Mean pain score (0–10) Sample 1: 7.1 (±1.3) Sample 2: 6.7 (±1.3)	Clinical setting (unspecified)	United States of America	English	-
	19	20	50.3 (29–64)	80%	Fibromyalgia	8.9 (−1–18)	Pain level (0–10) (SD) 6 (1.6)	Community	United States of America	English	
Medical Outcomes Study Sleep Scale	27	Sample 1 = 1011 Sample 2 = 173	Sample 1: 46 (18–94 range) Sample 2: 72 (31–100 range)	Sample 1: 51% Sample 2: 53%	Sample 1: Healthy subjects Sample 2: Postherpetic neuralgia	Sample 1: -Sample 2: 33.8 months (35.9)	-	Clinical Setting (unspecified)	United States of America	English	Sample 1: - Sample 2: Test: 100% Re-test: 51.44%
	19	20	50.3 (29–64)	80%	Fibromyalgia	8.9 (−1–18)	Pain level (0–10) (SD) 6 (1.6)	Community	United States of America	English	
	20	Sample 1: 748 Sample 2: 745	Sample 1: 48.8 (±10.9) Sample 2: 50.1 (±11.4)	Sample 1: 94.4% Sample 2: 94.5%	Fibromyalgia	Sample 1: ∼9 years Sample 2: ∼10 years	Mean pain score (0–10) Sample 1: 7.1 (±1.3) Sample 2: 6.7 (±1.3)	Clinical setting (unspecified)	United States of America	English	-
	21	129	49.4 (±11.0)	91.3%	Fibromyalgia	≥2 years	Moderate-to-severe in 88.1% of the sample	Community	United States of America	English	100%
Fibromyalgia Sleep Diary	24	FM experts = 4 FM patients = 34	FM patients: 47.8 (±11.9)	FM patients: 88.2%	Fibromyalgia	Not reported	Not reported	Community-based clinical sites	United States of America	English	100%

**Table 5 ijerph-17-02992-t005:** Results of studies on measurement properties (PSQI).

PROM (Ref)	Country (Language) in Which the PROM Was Evaluated	Internal Consistency			Test–Retest Reliability			Hypotheses Testing		
		n	Meth Qual	Result (Rating)	n	Meth Qual	Result (Rating)	n	Meth Qual	Result (Rating)
Pittsburgh Sleep Quality Index [17]	Spain (Spanish)	138	+	α = 0.805	96	+	ρ = 0.806 for the PSQI total score (*p* < 0.001). Lowest value ρ = 0.356 “daytime dysfunction” Highest value ρ = 0.718 “use of sleeping medication”	96	+	FIQ (total score) ρ = 0.304 (*p* < 0.01) SF-36 Physical functioning ρ = −0.372 (*p* < 0.01) Role physical ρ = −0.217 (*p* < 0.05) Role emotional ρ = −0.254 (*p* < 0.01) Vitality ρ = −0.247 (*p* < 0.05) Mental Health ρ = −0.208 (*p* < 0.05) Social functioning ρ = −0.426 (*p* < 0.01) Bodily pain ρ = −0.351 (*p* < 0.01) General Health NS
Pooled or summary result (overall rating)		138		0.805	96		0.806	96		FIQ: ρ = 0.304 (*p* < 0.01) SF-36: General Health NS Social functioning ρ = −0.426 (*p* < 0.01)

KMO: Kaiser-Meyer-Olkin, ρ: Spearman’s rank correlation coefficient, FIQ: Fibromyalgia Impact Questionnaire, SF-36: Short-Form health survey-36, NS: nonsignificant.

**Table 6 ijerph-17-02992-t006:** Results of studies on measurement properties (JSS).

PROM (Ref)	Country (Language) in Which the PROM Was Evaluated	Internal Consistency			Criterion validity			Reliability			Responsiveness		
		n	Meth Qual	Result (Rating)	n	Meth Qual	Result (Rating)	n	Meth Qual	Result (Rating)	n	Meth Qual	Results (Rating)
Jenkins Sleep Scale [18]	United States of America (English)	195	+	α = 0.70	195	+/−	FIQ item 16 r = 0.68 FIQ item 17 r = 0.72 Pain VAS r = 54 Fatigue VAS r = 57 ESS r = 0.43 FOSQ total score r = −0.57 SF-36 Vitality score r = −0.66	195	+/−	FIQ total score ICC 0.70 FIQ item 17 ICC 0.72 ESS ICC 0.69 Fatigue VAS ICC 0.66 Pain VAS ICC 0.61	R: 38 NR: 115	+	R: Pain VAS + FIQ total score SES = 1.62 NR Pain VAS + FIQ total score SES = −1.33
Pooled or summary result (overall rating)		195		0.70			0.43–0.72	195		0.61–0.72	R: 38 NR: 115		R: 1.62 NR: −1.33

r: Pearson Correlation Coefficient, ICC: Intraclass Correlation, α: Cronbach’s alpha, SES: Standardized effect sizes, FIQ: Fibromyalgia Impact Questionnaire, FOSQ: Functional Outcomes of Sleep Questionnaire, SF-36: Short-Form health survey-36, ESS: Epworth Sleepiness Scale, VAS: Visual Analogue Scale, R: Responders, NR: Non-responders.

**Table 7 ijerph-17-02992-t007:** Results of studies on measurement properties (SQ-NRS).

PROM (Ref)	Country (Language) in Which the PROM Was Evaluated	Criterion Validity			Test–Retest Reliability			Responsiveness		
		n	Meth Qual	Result (Rating)	n	Meth Qual	Result (Rating)	n	Meth Qual	Result (Rating)
Sleep Quality Numeric Rating Scale [20]	United States of America (English)	Sample 1 = 748 Sample 2 = 745	+	PNRS Sample 1 r = 0.64, *p* < 0.001 Sample 2 r = 0.58, *p* < 0.001 MOS-SS Sample 1 Sleep disturbance r = 0.45, *p* < 0.001 Snoring r = 0.01, *p* = 0.884 Awaken Short of breath of with headache r = 0.21, *p* < 0.001 Quantity of sleep r = −0.31, *p* < 0.001 Sleep adequacy r = −0.21, *p* < 0.001 Somnolence r = 0.11, *p* = 0.004 Sample 2 Sleep disturbance r = 0.42, *p* < 0.001 Snoring r = 0.00, *p* = 0.993 Awaken Short of breath of with headache r = 0.14, *p* < 0.001 Quantity of sleep r = −0.34, *p* < 0.001 Sleep adequacy r = −0.32, *p* < 0.001 Somnolence r = 0.15, *p* < 0.001	Sample 1 = 748 Sample 2 = 745	+	Sample 1 ICC 0.90 Sample 2 ICC 0.91	Pregabalin treatment Sample 1: 300 mg (n = 368) Sample 2: 450 mg (n = 373) Sample 3: 600 mg (n = 378)	+	Sample 1: SES = 0.46–0.52 Sample 2: SES = 0.59 Sample 3: SES = 0.73
Pooled or summary result (overall rating)		1493		PNRS 0.58–0.64 MOS-SS 0.00–0.45	1493		0.90–0.91			0.46–0.73

PNRS: Pain Numerical Rating Scale, MOS-SS: Medical Outcomes Measures-Sleep Scale, r: Pearson Correlation Coefficient, ICC: Intraclass Correlation.

**Table 8 ijerph-17-02992-t008:** Results of studies on measurement properties (MOS-SS).

PROM (Ref)	Country (Language) in Which the PROM Was Evaluated	Structural Validity			Internal Consistency			Test–Retest Reliability		
		n	Meth Qual	Result (Rating)	n	Meth Qual	Result (Rating)	n	Meth Qual	Result (Rating)
Medical Outcomes Study Sleep Scale [20]	United States of America (English)	Sample 1 = 748 Sample 2 = 745	−	CFA Bentler’s comparative fit index Baseline: 0.88 Week 5: 0.93 Week 9: 0.91 Week 13: 0.92	Sample 1 = 748 Sample 2 = 745	+/−	Sample 1: Week 1/week 13 Sleep disturbance subscale α = 0.78/α = 0.87 Somnolence subscale α = 0.72/α = 0.86 Sleep adequacy subscale α = 0.36/α = 0.74 Sample 2: Week 1/week 13 Sleep disturbance subscale α = 0.80/α = 0.87 Somnolence subscale α = 0.71/α = 0.75 Sleep adequacy subscale α = 0.61/α = 0.74			
Medical Outcomes Study Sleep Scale [21]	United States of America (English)							140	+	Week 1 = ICC 0.81Week 4 = ICC 0.89
Pooled or summary result (overall rating)		1493		0.88–0.93	1493		0.36–0.87	140		0.81–0.89

CFA: Confirmatory Factor Analysis, α: Cronbach’s alpha, ICC: Intraclass Correlation.

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
