# Peer review of "Patient Reported Outcome Measures of Sleep Quality in Fibromyalgia: A COSMIN Systematic Review"

_ijerph, 2020, doi:10.3390/ijerph17092992_

Round 1
Reviewer 1 Report
This systematic review is interesting for clinical audience, due to the necessity to evaluate sleep complains quickly with accuracy. the authors presented very well their findings and they have well justify their arguments related to each tool. I recommend his publication after minor clarifications in the discussion.
1) After reading the discussion, i was not able to understand what is the final recommandation about the tool/scale to use. Authors in the last paragraph in the same time suggest to develop an evaluation of sleep based on population with FM & to use validated measures of sleep already existing. What is the final conclusion? here below two studies i suggest to use and cite for this purpose:
https://www.ncbi.nlm.nih.gov/pmc/articles/PMC3691959/
https://www.ncbi.nlm.nih.gov/pmc/articles/PMC5447206/
2) I am wondering if in the discussion, authors can take another disease where sleep is evaluated with tools developed for/with another population/purpose compare with their findings in FM; to show potential challenges expected in future research. As an example, i suggest to use and cite the study below where a sleep questionnaire developed for general population was used in a population of gamers. Validity, reliability and findings were similar in this population with gaming disorder compared to general population:
https://www.nature.com/articles/s41598-020-58462-0
Author Response
Dear Reviewer,
We are pleased to resubmit for your consideration the revised version of the manuscript entitled " Patient Reported Outcome Measures of Sleep Quality in Fibromyalgia: A COSMIN Systematic Review".
We are very grateful for your excellent suggestions and comments.
We have carefully considered your suggestions and comments, and made revisions accordingly.
Thanks again for those valuable recommendations, which have helped us a lot in further improving the quality and clarity of this manuscript.
We have addressed each of the reviewers´ requirements as outlined below and are indicated in the manuscript in green color:
REVIEWER 1
This systematic review is interesting for clinical audience, due to the necessity to evaluate sleep complains quickly with accuracy. the authors presented very well their findings and they have well justified their arguments related to each tool. I recommend his publication after minor clarifications in the discussion.
Comments:
- After reading the discussion, i was not able to understand what is the final recommendation about the tool/scale to use. Authors in the last paragraph in the same time suggest to develop an evaluation of sleep based on population with FM & to use validated measures of sleep already existing. What is the final conclusion? here below two studies i suggest to use and cite for this purpose:
Thank you for your comment, we have added a new sentence discussing the extensive use of the PSQI in the existing literature and citing the suggested articles. Please, see page 11 and below:
“In this regard, the PSQI is the most widely used PROM among the existing literature in the field of FM providing relevant information about both the relationship of poor sleep quality to other symptoms of this health condition [30-33] and the effects of different treatment approaches [34-37].”
- I am wondering if in the discussion, authors can take another disease where sleep is evaluated with tools developed for/with another population/purpose compare with their findings in FM; to show potential challenges expected in future research. As an example, i suggest to use and cite the study below where a sleep questionnaire developed for general population was used in a population of gamers. Validity, reliability and findings were similar in this population with gaming disorder compared to general population: https://www.nature.com/articles/s41598-020-58462-0
We appreciate the reviewer’s suggestion. Although the recommended study is interesting, we decided not to include it in the discussion as the topic is out of the scope of the present study.
In the name of the authors, I hope that this revised manuscript answers all the concerns contained in the reviews and we are grateful for the thought and effort the reviewers have put into these reviews.
Yours sincerely.
Joan Blanco-Blanco
Reviewer 2 Report
Patient Reported Outcome Measures of Sleep Quality in 3 Fibromyalgia: A COSMIN Systematic Review
The authors present a systematic review of the patient reported outcome measures of sleep quality validated in adult people diagnosed with fibromyalgia. A total of five PROMS where included in the analysis all of them rated with low risk of bias according with COSMIN criteria. All of them were found reliable and valid in some psychometric properties but none of them included all the psychometric properties stated in the COSMIN guidelines.
The authors performed a detailed literature search, assessed the quality of the results based on a COSMIN checklist for risk bias and selected publications that met the criteria presenting a PRISMA flow diagram. They described characteristics, evaluated the content validity, internal structure and measurement properties available for each instrument. In addition, they mentioned if there were adaptations or translations of the measurement in patients with fibromyalgia.
They conclude that the five measurements found are valid and reliable, nevertheless just one of them has the complete list of psychometric properties established in the COSMIN guidelines.
The overall paper would be benefit from a slight English language review, methods are well performed and explained and results well presented. I would suggest a more specific conclusion regarding the quality of just one instrument.
Comments:
Confused idea or mistake in Figure 1 on the Eligibility section, the second text box explains the 6 full abstracts excluded, but the number followed 7 studies, it doesn’t make sense to me, is there a mistake?
Line 193-198 PSQI compositions is exposed as 19 items but it fails to explain clearly how they are distributed along the index.
Line 229 I suggest adding that the group III was composed of 62 patients with sleep disorders, as the original article cites.
302 sleep word is repeated
336 check the spelling
Table 5-7 would be benefit with a better presentation for readers.
Author Response
Dear Reviewer,
We are pleased to resubmit for your consideration the revised version of the manuscript entitled " Patient Reported Outcome Measures of Sleep Quality in Fibromyalgia: A COSMIN Systematic Review".
We are very grateful for your excellent suggestions and comments. We have carefully considered all the suggestions and comments, and made revisions accordingly.
Thanks again for those valuable recommendations, which have helped us a lot in further improving the quality and clarity of this manuscript. We have addressed each of the reviewers´ requirements as outlined below and are indicated in the manuscript in green color.
The authors noticed that the lines in the document tha you revised are numbered differently compared to the manuscript that we originally sent. For example, you mentioned a mistake in line 302 but in our document this line corresponds with line 287. However, we were able to find the fragments of the text where you made the suggestions and carried out the corresponding amendments:
REVIEWER 2
Patient Reported Outcome Measures of Sleep Quality in 3 Fibromyalgia: A COSMIN Systematic Review
The authors present a systematic review of the patient reported outcome measures of sleep quality validated in adult people diagnosed with fibromyalgia. A total of five PROMS where included in the analysis all of them rated with low risk of bias according with COSMIN criteria. All of them were found reliable and valid in some psychometric properties but none of them included all the psychometric properties stated in the COSMIN guidelines.
The authors performed a detailed literature search, assessed the quality of the results based on a COSMIN checklist for risk bias and selected publications that met the criteria presenting a PRISMA flow diagram. They described characteristics, evaluated the content validity, internal structure and measurement properties available for each instrument. In addition, they mentioned if there were adaptations or translations of the measurement in patients with fibromyalgia.
They conclude that the five measurements found are valid and reliable, nevertheless just one of them has the complete list of psychometric properties established in the COSMIN guidelines.
The overall paper would be benefit from a slight English language review, methods are well performed and explained and results well presented. I would suggest a more specific conclusion regarding the quality of just one instrument.
Comments:
- Confused idea or mistake in Figure 1 on the Eligibility section, the second text box explains the 6 full abstracts excluded, but the number followed 7 studies, it doesn’t make sense to me, is there a mistake?
Thank you for your observation. There is a mistake in the box as the number of studies excluded for being meeting abstracts is 4 and not 5. We have modified the numbers. Please, see the box on page 20 and bellow:
“Full-text articles excluded, with reasons” of the Figure 1. PRISMA Flow diagram.
- Line 193-198 PSQI compositions is exposed as 19 items but it fails to explain clearly how they are distributed along the index.
The authors have noticed that the lines in the document of the reviewer are numbered differently compared to the manuscript that we originally send. The lines 193-198 for the reviewer correspond with lines 178-183 for the authors.
The authors appreciate the reviewer’s comment. We have detailed the 7 components of the PSQI in lines 189-191. However, we agree with the reviewer that this information should have been presented in the reviewer’s recommended section. We have amended the mistake by moving the information from lines 189-191 to lines 192-195 (according to the reviewers numbered lines). Please, see pages 192-195 and below:
“The items of the PSQI are organized into 7 components: 1) Subjective sleep quality, 2) Sleep latency, 3) Sleep duration, 4) Habitual sleep efficiency, 5) Sleep disturbances, Use of sleeping medication, and 7) daytime dysfunction.”
- Line 229 I suggest adding that the group III was composed of 62 patients with sleep disorders, as the original article cites.
Thank you for your comment. We have added the lacking information about the sample characteristics. Please, see line 229 and below:
[…] 62 patients with sleep disorders referred by a physician from another psychiatric center formed group III.
- 302 sleep word is repeated
Thank you for your observation. We have deleted the repeated word “sleep”. Please, see page 8 and below:
“Objective of the tool: The MOS-SS allows the assessment of sleep quality.”
- 336 check the spelling
We appreciate your observation. As indicated, we have corrected the misspelling. Please, see line 336 and below:
“[…] How difficult was it to […]”
- Table 5-7 would be benefit with a better presentation for readers.
We appreciate your comment. The tables are presented in accordance with the COSMIN templates provided in the COSMIN web-site “Help extracting data and reporting all your results”
https://www.cosmin.nl/tools/guideline-conducting-systematic-review-outcome-measures/
However, we have changed the design of the table so as to facilitate the reading of the data.
In the name of the authors, I hope that this revised manuscript answers all the concerns contained in the reviews and we are grateful for the thought and effort the reviewers have put into these reviews.
Yours sincerely.
Joan Blanco-Blanco